

**Tree height uncertainty biases aboveground biomass estimation more**
**than wood density in miombo woodlands**
Arthur M. Yambayamba[1,2*], Ferdinand Handavu[3], Kondwani Kapinga[4,5], Tommaso Jucker[1]
**Affiliations**
[1]School of Biological Sciences, University of Bristol, Bristol, UK
[2]School of Natural Resources, Copperbelt University, Kitwe, Zambia
[3]Department of Geography, Environment and Climate Change, Mukuba University, Kitwe,
Zambia
[4]Dag Hammarskjöld Institute for Peace and Conflict Studies (DHIPS), Environment,
Sustainable Development and Peace, Copperbelt University, Kitwe, Zambia
[5]Chair of Environment and Development, Oliver R. Tambo Africa Research Chair Initiative
(ORTARChI), Copperbelt University, P.O. Box 21692, Kitwe, Zambia
**Corresponding author**
Arthur M. Yambayamba, School of Biological Sciences, University of Bristol, 24 Tyndall
Avenue, Bristol, BS8 1TQ, UK.
Email: arthur.yambayamba@bristol.ac.uk





**ABSTRACT**
Accurate and unbiased estimation of tree aboveground biomass ($AGB$) is essential for large-
scale monitoring of forest carbon stocks. But estimating $AGB$ typically requires several data
imputation steps that can introduce substantial errors that are hard to quantify and correct for.
Two sources of uncertainty that are thought to be particularly important but remain poorly
understood are tree height – which is generally estimated using allometric models – and wood
density – which is most commonly assigned from databases based on taxonomic matching.
Here we used data from 154 destructively harvested trees in Zambia's miombo woodlands that
span a large range of sizes to develop a framework to partition errors in $AGB$ arising from
uncertainty in tree height and wood density. We found that when locally-calibrated allometries
are used to estimate missing tree height information and when wood density is imputed from
species-specific values derived from public databases, $AGB$ can be estimated with high
precision and little or no bias. However, when tree height and wood density are imputed more
coarsely using generic information, errors in $AGB$ can be substantial. In particular, estimating
tree height using a regional allometric model developed for tropical dry forests led to 35%
underestimation of $AGB$. Our study provides an intuitive approach for quantifying and
partitioning errors in $AGB$ arising from uncertainty in tree height and wood density, paving the
way for more robust mapping of forest carbon stocks and fluxes.
**Key words**: Aboveground biomass, allometry, bias, error propagation, forest carbon stocks,
forest structure, tree height, wood density



## 1. INTRODUCTION

Forests and savannas play a crucial role in regulating the terrestrial carbon cycle by sequestering large amounts of carbon dioxide ($CO_2$) from the atmosphere and storing it as woody biomass (Chave et al., 2014; Rathgeber et al., 2016; Mitchard, 2018; Martin et al., 2018). This potential of forests to act as long-term carbon sinks, that slow the pace of climate change by partially offsetting anthropogenic $CO_2$ emissions, has gained widespread attention in recent decades (Hansen et al., 2013; Ipcc, 2013; Potapov et al., 2017; Grantham et al., 2020; Ellis et al., 2021; Arias et al., 2021). This has led to growing interest in restoration programs centered around forest creation and expansion (Lewis et al., 2019; Hua et al., 2024; Cheng et al., 2024), as well as burgeoning international forest carbon markets underpinned by programs such as REDD+ (Kalaba et al., 2013; Bomfim et al., 2022). However, a fundamental assumption of these efforts is that forest carbon stocks can be estimated accurately and without bias. In practice, this is rarely the case even at the most basic level of the individual tree (Demol et al., 2024; Fareed and Numata, 2024; Terryn et al., 2024), as there are multiple sources of uncertainty that affect tree biomass estimation and we lack a clear understanding of their relative importance and magnitude.

A fundamental unit of forest carbon stock estimation is a tree's aboveground woody biomass ($AGB$). Direct measurements of $AGB$ involve destructive sampling to weigh the trunk and branches, which in addition to killing the tree is laborious and costly (Clark and Kellner, 2012; Colgan et al., 2013). In practice, $AGB$ is therefore almost always estimated using allometric equations that have been calibrated using data from relatively small numbers of trees whose mass has been destructively measured (Mugasha et al., 2013; Ngomanda et al., 2014; Kapinga et al., 2018; Handavu et al., 2021) – although more recently terrestrial laser scanning is being increasingly used to generate accurate, non-destructive estimates of tree woody volume and $AGB$ (Momo Takoudjou et al., 2018; Calders et al., 2022; Demol et al., 2024). These



allometric biomass equations typically have some combination of stem diameter at breast
height ($D$), tree height ($H$), wood density ($\rho$) and, in rarer cases, crown width as input variables
(Chave et al., 2005; Chave et al., 2014; Goodman et al., 2014; Sileshi, 2014; Ploton et al., 2016;
Jucker et al., 2017). A myriad of such equations exist, but the most widely used for tropical
trees are the models developed by (Chave et al., 2005; Chave et al., 2014), which express $AGB$
as a power-law function of $D^2{\times}H{\times}\rho$ – a compound variable that approximates a tree's trunk
volume ($D^2{\times}H$) and multiplies this by the density of its wood to estimate $AGB$.

69         These pantropical $AGB$ equations generally perform very well when compared to data

from destructive harvests. But when measured inputs are replaced with estimates derived from
forest inventories, several important sources of bias and uncertainty can be introduced. In
particular, while $D$ is generally recorded accurately and precisely in forest inventories, the same
is not true for $H$ and $\rho$. Common approaches to measuring $H$ in the field using clinometers and
laser range finders are both difficult to do accurately and time consuming. When $H$ is measured
directly in the field, estimates are therefore often uncertain and in some cases systematically
biased (Larjavaara and Muller-Landau, 2013; Sullivan et al., 2018; Terryn et al., 2024). More
commonly, $H$ is not measured at all, but is instead itself estimated from $D$ using allometric
equations (Fayolle et al., 2016; Sullivan et al., 2018; Kafuti et al., 2022). These $H$–$D$ allometries
not only propagate any underlying biases in the $H$ values used to derive them, but can also
introduce additional sources of uncertainty that are hard to quantify once scaled to $AGB$. This
includes errors arising from the choice of functional form used to model $H$–$D$ allometries
(Fayolle et al., 2016; Ledo et al., 2016; Cano et al., 2019; Terryn et al., 2024) and the fact that
$H$–$D$ scaling relationships can vary considerably among tree species and forest types
(Feldpausch et al., 2011; Banin et al., 2012; Jucker et al., 2022; Jucker et al., 2025) – something
that most $H$–$D$ models completely overlook.



86       Similarly, $\rho$ is very rarely measured as part of field inventories and is instead almost

always assigned from global databases based on taxonomic or geographic matching (Chave et
al., 2009; Réjou-Méchain et al., 2017). While these databases are invaluable, they still only
cover <15% of known tree species, meaning that in many cases $\rho$ values are imputed using
either taxonomic (e.g., genus or family-level means) or geographic proxies (e.g., plot or
regional-level means). Depending on a region's taxonomic coverage in the database and the
choice of imputation method, uncertainty in $\rho$ can therefore substantially bias $AGB$ estimates
(Flores and Coomes, 2011; Mitchard et al., 2014; Phillips et al., 2019).

94       Here we provide a rigorous quantitative assessment of how uncertainty in both $H$ and

$\rho$ propagates to tree-level $AGB$ estimates, focusing specifically on southern Africa's miombo
woodlands. Like other tropical dry forests, miombo woodlands remain underrepresented in the
research and conservation agenda despite covering an area approximately ten times the size of
the UK and storing substantial amounts of carbon in their vegetation (Pennington et al., 2018;
Mcnicol et al., 2018; Moonlight et al., 2021; Demol et al., 2024). To address this knowledge
gap, we compiled data for 154 trees whose $AGB$ was measured via destructive harvesting in
miombo woodlands of Zambia. These trees cover a wide spectrum of sizes ($D$ = 5.0–52.2 cm;
$H$ = 3.0–25.0 m) and represent 37 uniquely identified species that vary considerably in their
wood density ($\rho$ = 0.34–0.86 g cm$^{-3}$). Using these data, we set out to address two key research
objectives. First, we compared $AGB$ estimates obtained using new biomass allometric models
fit to the data with those of existing local and pantropical biomass models. Second, we
systematically assessed how different approaches to imputing $H$ and $\rho$ affect tree-level $AGB$
estimates.



## 2. MATERIALS AND METHODS

### 2.1 Study system

The study was conducted using multiple datasets acquired in Zambia's miombo woodlands. These tropical dry forests are characterized by the dominance of trees in the genus *Brachystegia* that form open-canopy habitats where trees and grasses co-exist. Regionally, they cover around 2 million km$^2$ of land, forming a belt that stretches from Angola's Atlantic coast to the Indian Ocean in Mozambique and Tanzania (Ribeiro et al., 2015; Ryan et al., 2016; Dziba et al., 2020). Miombo woodlands are shaped by long dry seasons followed by months of intense rainfall, with fire playing a key role in keeping the balance between trees and grasses. Trees in this region have adapted to take advantage of short growing seasons and are able to recover remarkably quickly from wildfires and other forms of disturbance, allowing these woodlands to store surprisingly large amounts of carbon in their vegetation given the seasonally dry climate (Mcnicol et al., 2018; Pelletier et al., 2018; Demol et al., 2024).

### 2.2 Tree harvest data

We compiled three separate datasets where individual trees were destructively harvested to measure their aboveground biomass ($AGB$, in kg) (Kapinga et al., 2018; Handavu et al., 2021). This involved felling the trees at ground level and then cutting their stems, branches and twigs into sections so that these could be weighed to estimate the total aboveground woody mass of each tree. In addition to $AGB$, the stem diameter at breast height ($D$, in cm), tree height ($H$, in m), and wood density ($\rho$, in g cm$^{-3}$) of each tree was also measured. A total of 37 uniquely identified tree species spanning a broad range of wood densities (0.34–0.86 g cm$^{-3}$; median = 0.61 g cm$^{-3}$) were measured (Table 1). Together, the harvested trees covered a large spectrum of sizes ($D$ = 5.0–52.2 cm; $H$ = 3.0–25.0 m), with $AGB$ ranging between 5.7–2374.4 kg (Table



1). *AGB* values were visually assessed for potential data entry errors and four trees were flagged
as outliers and excluded (Fig. S1a). This left us with a total of 154 trees for fitting *AGB* models.
**Table 1.** Breakdown of the tree harvest dataset for the three sites in miombo woodlands,
including number of trees harvested and species, as well as the median and range (in square
brackets) of stem diameter at breast height, total tree height, wood density, aboveground
biomass, elevation, mean total annual precipitation and mean annual temperature.

| | Katanino forest | Miengwe forest | Mwekera forest | Combined dataset |
|---|---|---|---|---|
| Number of trees harvested | 12 | 94 | 48 | 154 |
| Number of species | 9 | 33 | 5 | 37 |
| Stem diameter (*D*; cm) | 22.6 [12.2 – 46.0] | 21.3 [5.0 – 52.2] | 22.0 [5.5 – 48.0] | 21.4 [5.0 – 52.2] |
| Tree height (*H*; m) | 13.5 [5.7 – 22.8] | 12.9 [4.2 – 20.7] | 16.5 [5.0 – 25.0] | 13.8 [4.2 – 25.0] |
| Wood density ($\rho$; g cm$^{-3}$) | 0.64 [0.49 – 0.86] | 0.60 [0.42 – 0.82] | 0.64 [0.34 – 0.85] | 0.61 [0.34 – 0.86] |
| Aboveground biomass (*AGB*; kg) | 301.0 [27.2 - 2149.3] | 217.7 [7.8 - 2374.4] | 225 [5.7 – 1545.2] | 218.9 [5.7 – 2374.4] |
| Mean elevation (m a.s.l.) | 1323 | 1324 | 1228 | [1228 – 1324] |
| Mean annual precipitation (mm yr$^{-1}$) | 1325 | 1250 | 1198 | [1198 – 1325] |
| Mean annual temperature (°C) | 20.1 | 20.2 | 20.9 | [20.1 – 20.9] |






### 2.3 Height–diameter modelling data

To determine how different approaches to impute missing tree height values impact *AGB* estimates, we compiled an additional dataset of trees from miombo woodlands in Zambia for which both *H* and *D* were measured in the field. This included forest inventory data from the three sites where trees were also destructively harvested to measure *AGB* (Kapinga et al., 2018; Handavu et al., 2021). Collectively, these data represent 79 uniquely identified tree species and span a similar range of tree sizes to those sampled for *AGB* (*D* = 5.0–70 cm; *H* = 2.0–33.0 m) (see Table S3 for details). *H–D* relationships were visually assessed for potential data entry errors and a small number of trees (~2%) were flagged as outliers and excluded (Fig. S1b). This left us with a total of 4321 trees for model fitting.

### 2.4 Data harmonization and analysis

All data processing and statistical analyses were conducted in R (version 4.3.1; (R Core Team, 2023)) using the brms, GGally, terra, tidyverse, data.table, ggpmisc, cowplot, multcompView, BIOMASS, rWCVP and U.Taxonstand packages (Bürkner, 2017; Réjou-Méchain et al., 2017; Wickham et al., 2019; Wilke, 2020; Schloerke et al., 2021; Hijmans, 2023; Dowle and Srinivasan, 2023; Brown et al., 2023; Zhang and Qian, 2023; Aphalo, 2024). Prior to conducting statistical analyses, tree species names were harmonized across all datasets using the World Checklist of Vascular Plants (Govaerts et al., 2021; Brown et al., 2023). For this step, we also cross-referenced and matched species names against the Global Wood Density Database (GWDD) which we used in subsequent analyses to assign $\rho$ values to species (Chave et al., 2009; Réjou-Méchain et al., 2017).

### 2.5 Comparing alternative *AGB* models

To determine how much *AGB* estimates are influenced by the choice of biomass allometry, we compared *AGB* estimates obtained using two existing and two newly-developed biomass





models. The first $AGB$ allometry we tested is the widely used pantropical model developed by
(Chave et al., 2014), which expresses $AGB$ as the following function of $D$, $H$ and $\rho$:

$$AGB = 0.067 \times (\rho D^2 H)^{0.976} \times \exp\left[\frac{0.357^2}{2}\right]. \tag{1}$$

The second is a local $AGB$ model developed as part of Zambia's second integrated land-
use assessment (ILUA2) which estimates $AGB$ from $D$ alone (Ilua, 2016; Forestry Department,

166     2016):

$$AGB = 0.128 \times D^{2.342}. \tag{2}$$

To complement these two existing $AGB$ models, we used the tree harvest data described
above to reparametrize a miombo-specific version of both equations. Specifically, we used the
brms package to fit log–log regression models in a Bayesian framework were $AGB$ was
expressed as a function of either $\rho D^2 H$ or $D$ alone (see S1 for details on model fitting). The
resulting fitted models were as follows:

$$AGB = 0.118 \times (\rho D^2 H)^{0.924} \times \exp\left[\frac{0.373^2}{2}\right], \text{ and} \tag{3}$$

$$AGB = 0.081 \times D^{2.600} \times \exp\left[\frac{0.385^2}{2}\right]. \tag{4}$$

We then compared the predictive ability of these four models on the basis of two widely
used metrics that capture both the precision and bias of model predictions relative to observed
$AGB$ values (Huang et al., 2003; Chave et al., 2014; Sileshi, 2014): root mean square error
(RMSE, in kg) and percentage error (PE, in %):
Root mean square error (RMSE) = $\sqrt{\left[\frac{1}{n}\sum_{i=1}^{n}\left(AGB_i - \widehat{AGB_i}\right)^2\right]}$, and $\qquad$ (5)
Percentage error (PE) = $\left(\left(\frac{1}{n}\sum_{i=1}^{n}\left(\widehat{AGB_i} - AGB_i\right)\right) \Big/ \frac{1}{n}\sum_{i=1}^{n}(AGB_i)\right) \times 100$, $\qquad$ (6)
where $AGB_i$ is the measured $AGB$ for tree $i$ and $\widehat{AGB_i}$ is the predicted $AGB$ for tree $i$.



We used PE in order to get the magnitude of the mean error relative to the mean
observed *AGB*. We chose PE among other metrics because it gives much more weight to the
large trees since these have the largest errors in absolute terms. Large errors are undesirable for
large trees because these contribute the most to total *AGB* (Fig. 1a).
Finally, we also assessed the performance of the models by fitting a linear regression
between observed and predicted *AGB* values and extracting the intercept and slope parameters
with their associated 95% confidence intervals. If the model is unbiased, we would expect the
intercept to have a value close to 0 and the slope to be approximately 1 (Sileshi, 2014).







**Figure 1.** Overview of the tree harvest dataset. The bar plot **(a)** shows the distribution of

samples across the five tree diameter classes in terms of both number of trees and their



contribution to above ground biomass (*AGB*), both expressed as percentages of the total.
Scatterplots show the relationship between *AGB* and the predictors used to model it, including
**(b)** stem diameter (*D*), **(c)** total tree height (*H*), **(d)** wood density ($\rho$) and **(e)** the compound
variable of *D*, *H* and $\rho$. The coefficient of determination ($R^2$) between *AGB* and each of the
predictor variables is shown along with a line of best fit (blue colour) and its 95% confidence
bound (grey colour). The dark grey filled circles represent the *AGB* values of each of the 154
destructively harvested trees, plotted on a log-log scale.














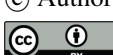



**2.6 Influence of height and wood density uncertainty on *AGB* estimates**


To better understand how different approaches to imputing $H$ and $\rho$ impact tree-level *AGB*
estimates, we devised a series of scenarios in which we imputed values of $H$ and $\rho$ following
commonly used practices for dealing with missing data and quantified their relative
contribution to *AGB* errors. For the purposes of this analysis, we used the pantropical model of
(Chave et al., 2014) to predict *AGB* (Eq. (1)), as it has been widely used in the tropics, including
in miombo woodlands (Pelletier et al., 2018; Grz, 2021; Kanja et al., 2025), and performed
well on our dataset (see Results for details). As above, model performance was assessed by
comparing predicted and observed *AGB* values on the basis of RMSE and PE for the entire
dataset, as well as PE when grouping trees into different $D$ size classes (<10 cm, 10–20 cm,
20–30 cm, 30–40 cm, >40 cm).
As a starting point, we began by using Eq. (1) to predict the *AGB* of all 154 harvested
trees when using field-measured values of both $H$ and $\rho$ as inputs. This served as our best case
scenario, in which all input data have been directly measured in the field. Next, we repeated
this same process but replaced field-measured $H$ values with estimates derived from two
alternative $H$–$D$ allometries: (1) a locally-calibrated, species-specific allometry and (2) a
regional, biome-specific allometry. The local allometry was derived using $H$ and $D$ values from
the 4321 trees described above. The relationship was modelled as a power-law function fit to
log–log transformed data, where both the scaling coefficient ($\alpha$, intercept) and scaling exponent
($\beta$, slope) were allowed to vary across species ($j$) in a hierarchical Bayesian framework (see S2
for details):

$$H = \alpha_j \times D^{\beta_j}. \tag{7}$$

For the regional $H$–$D$ allometry, we instead used an existing model for tropical dry forest trees
developed using the Tallo database (Jucker et al., 2022):



$$H = 2.355 \times D^{0.477}. \tag{8}$$

235  Finally, we used a similar approach to vary the wood density inputs in the *AGB*

236 equation. Specifically, we replaced field-measured $\rho$ values with either (1) species-specific

237 mean values (or closest taxonomic unit) obtained from the GWDD or (2) with a regional mean

238 value for African tropical forests ($\rho = 0.598$ g cm$^{-3}$; (Chave et al., 2009). When matching to the

239 GWDD, 55% of trees were assigned species-level values, 43% genus-level means and the

240 remaining 2% a mean of the population (0.682 g cm$^{-3}$). In total, this gave us nine scenarios to

241 compare: three possible *H* inputs into the *AGB* equation (field-measured, species-specific or

242 biome-specific), three possible wood density inputs (field-measured, species-specific or

243 biome-specific), and their respective combinations.















## 3. RESULTS

### 3.1 Uncertainty in *AGB* estimation due to choice of *AGB* model

Of the four *AGB* models we compared, three performed similarly well and produced unbiased estimates of *AGB* across the range of tree sizes (Fig. 2): the two models calibrated using the tree harvest data presented in this paper (Eq. (3) and Eq. (4)) and the existing pantropical model developed by Chave et al. (2014) (Eq. (1)). Regressions between predicted and observed *AGB* values for all three models had intercepts with 95% confidence intervals that overlapped with 0 (range of mean intercept values = -13.1 – 16.3) and slopes that were very close to 1. Of these three *AGB* models, the *D*-only model (Fig. 2c) had the lowest RMSE (160.5 kg vs 210.6 and 217.9 kg) and PE (0.6% vs -2.4 and -10.3%), confirming that *D* is the single strongest predictor of *AGB* (Fig. 1b). By contrast, the ILUA2 model (Eq. (2)) developed as part of Zambia's forest inventory program substantially underestimated *AGB* (Fig. 2a), especially for large trees (green line in Fig. 2f). On average, predicted values of *AGB* obtained using the ILUA2 model were around 40% lower than those measured in the field (slope coefficient = 1.8) and trees in the largest size class had a PE of -40% (Fig. 2f).





**Figure 2.** Predictive accuracy of the four alternative aboveground biomass (*AGB*) models.

Scatterplots (**a–d**) show the relationship between observed and predicted *AGB* values across

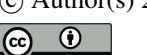


the 154 trees included in the dataset. Coloured lines correspond to the fit of a linear model
between observed and predicted $AGB$ values, while dashed lines illustrate a 1:1 relationship.
Fit statistics for each model are reported in the panels, including the intercept (a) and slope (b)
of the regression line with 95% confidence intervals in brackets. Model errors are visualised in
the bottom two panels, including **(e)** the distribution of relative errors, as a percentage,
(calculated as: $(\widehat{AGB}_i - AGB_i)/AGB_i) * 100;$ where $AGB_i$ is the measured $AGB$ for tree $i$;
and $\widehat{AGB}_i$ is the estimated $AGB$ for tree $i$) for each of the four models and **(f)** the percentage
error (PE, in %) of trees grouped into the five stem diameter classes.

















**3.2 Error in *AGB* estimation due to uncertainty in height and wood density**


Across the nine scenarios in which we modified height (*H*) and wood density (*ρ*) inputs in the
pantropical *AGB* model, we found that replacing field-measured values of *H* and *ρ* with species-
specific estimates did not substantially worsen *AGB* estimates (Figs 3–4). In particular, height
estimates obtained using the locally-calibrated, species-specific *H–D* allometry (Eq. (7)) were
statistically indistinguishable from those measured in the field (Fig. S5a) and consequently had
little impact on predicted *AGB* values (Fig. 3b). Wood density values obtained from the GWDD
were correlated (Pearson's correlation coefficient = 0.41; Fig. S7a) but generally higher than
those measured in the field (Figs. S5b), and consequently, their use in the *AGB* model did lead
to a modest reduction in underestimation of *AGB* (PE increase from -10.3% to -3.4%; Fig. 3a
and Fig. 3d). However, even in this case, the slope coefficient between observed and predicted
*AGB* values was almost exactly 1 and the intercept very close to 0 (Fig. 3d).
By contrast, when *H* and *ρ* values were replaced with coarser regional-level estimates,
errors in *AGB* increased substantially. This was most pronounced for scenarios that involved
using height estimates derived from the generic *H–D* allometry for tropical dry forests (Eq. (8);
right column of Fig. 3). Because this allometry severely underestimated tree heights by several
meters (Fig. 5 and Fig. S6 and Table S6), by itself its use led to a 35% underestimation of *AGB*
(slope coefficient between observed and predicted values = 1.6; Fig. 3c). A similar, albeit much
weaker trend was also observed for wood density. The regional mean *ρ* value for African
tropical forests was only slightly lower than the mean *ρ* value of species sampled in our study
(0.598 vs 0.607 g cm$^{-3}$). Therefore, replacing field-measured *ρ* values with a regional mean
resulted in a slight increase in underestimation of *AGB* from -10.3% to -15.7% (Fig. 3g; slope
coefficient = 1.2). When these two sources of errors were combined, they compounded each
other and led to an underestimation of *AGB* that was comparable in magnitude to that of using



the ILUA2 model to estimate *AGB* instead of the pantropical equation of Chave et al. (2014)
(Fig. 3i; slope coefficient = 1.7).

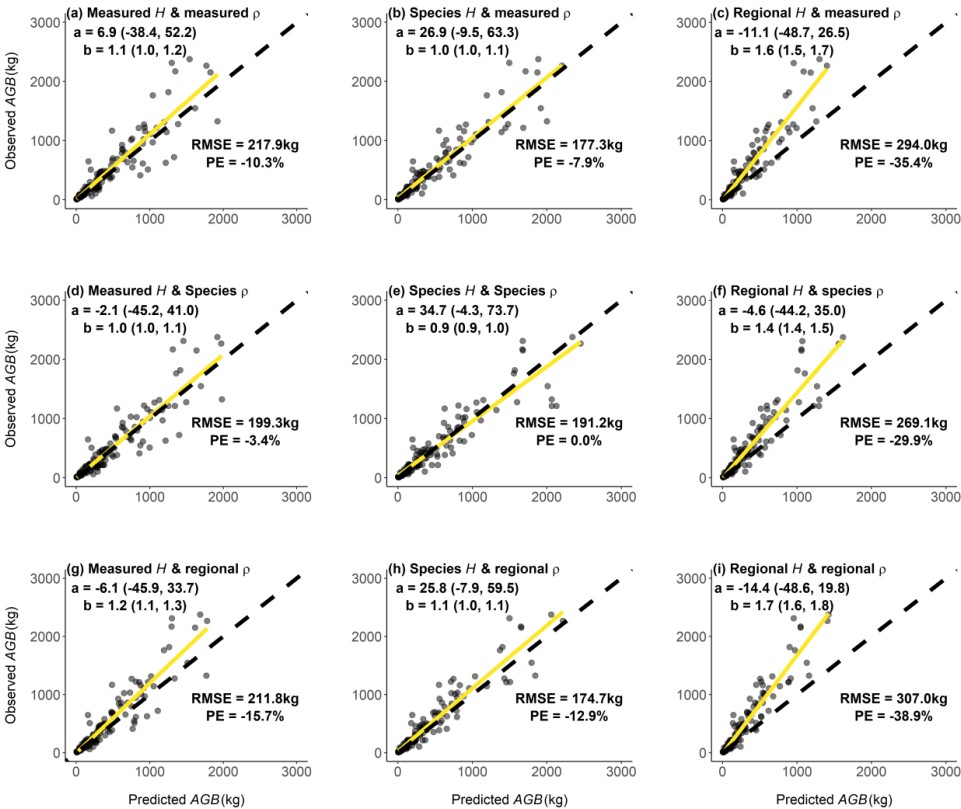


**Figure 3.** Errors in aboveground biomass (*AGB*) estimation arising from uncertainty in tree

height (*H*) and wood density ($\rho$). Each panel corresponds to one of the nine scenarios described

in the main text in which we altered inputs of *H* and $\rho$ in the *AGB* model developed by Chave

et al. (2014) (Eq. (1)). Panel (**a**) shows the predicted values of *AGB* when using field-measured

values of both *H* and $\rho$ (equivalent to Fig. 2b). From left to right, *H* values are replaced first

with species-specific estimates derived from the locally fit *H–D* allometry (middle column)

shown in Fig.5 and then with the regional *H–D* allometry for tropical dry forests derived from

the literature (right column) shown in Fig. 5. From top to bottom, $\rho$ values are replaced first



with species-specific mean values derived from the Global Wood Density Database (middle
row) and then with a single regional mean value for African tropical forests derived from the
literature (bottom row). Coloured lines correspond to the fit of a linear model between observed
and predicted $AGB$ values, while dashed lines illustrate a 1:1 relationship. Fit statistics for each
model are reported in the panels, including the intercept (a) and slope (b) of the regression line
with 95% confidence intervals in brackets.
































**Figure 4.** Errors in aboveground biomass ($AGB$) estimation due to uncertainty in tree height
($H$) and wood density ($\rho$) across the nine scenarios depicted in Fig. 3. Panels on the left show
the distribution of relative errors, as a percentage, (calculated as: $((\widehat{AGB_t} - AGB_i)/AGB_i) *$
$100$; where $AGB_i$ is the measured $AGB$ for tree $i$; and $\widehat{AGB_t}$ is the estimated $AGB$ for tree $i$)
in $AGB$ across the nine scenarios, while those on the right show the percentage error (PE, in
%) for trees grouped into the five stem diameter classes across scenarios. In all cases, $AGB$ was
estimated using the pan-tropical model developed by Chave et al. (2014) (Eq. (1)).

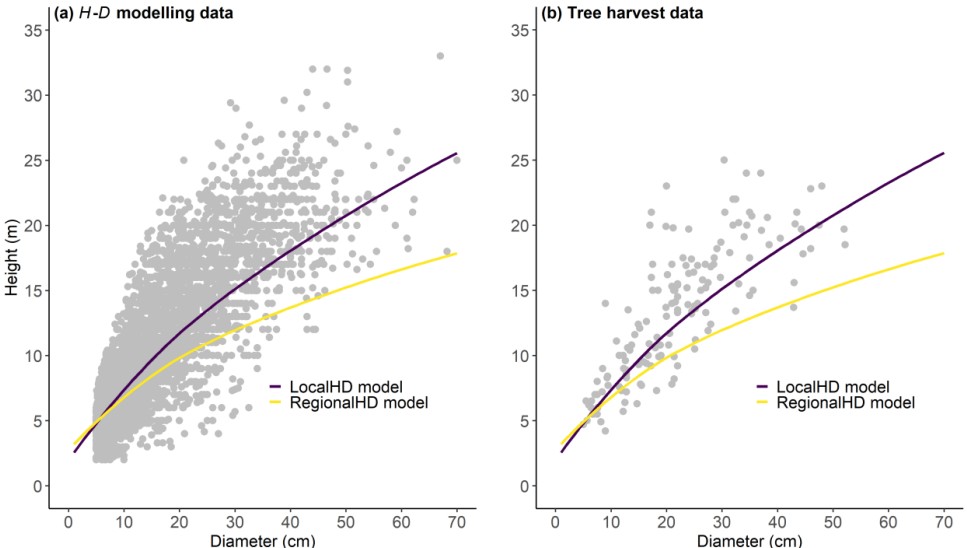


**Figure 5.** Relationship between tree height ($H$) and stem diameter ($D$) for **(a)** the 4321 trees
used to develop the species-specific, local $H$–$D$ allometry and **(b)** the 154 trees in the tree
harvest dataset used for $AGB$ modelling. The line of best-fit of the local power-law model fit
to these data (Eq. (7)) is shown in purple, while the yellow curve shows the fit of the regional
$H$–$D$ allometry for tropical dry forests derived from the literature (Eq. (8)). To visualise the
models, we generated a predicted $H$ for $D$ values in the range 1-70 cm (with 1cm interval) and
converted point predictions into lines via a locally estimated scatterplot smoothing (LOESS).





## 4. DISCUSSION

### 4.1 Strength in numbers: pantropical model provides robust estimates of *AGB*

Our study shows that when accurate measurements or estimates of tree height and wood density are available, the pantropical model developed by Chave et al. (2014) can generate unbiased estimates of *AGB*. In fact in our case, this pantropical model performed similarly well to *AGB* allometries calibrated against the data (Fig. 2). A key feature of this pantropical model is that compared to other regional or species-specific models that exist in the literature, it was developed using a large number of harvested trees (>4000) that span a broad range of environments and sizes. This is clearly a major strength when it comes to making predictions against new data. A priority for future research in this field should therefore be to further consolidate and expand data on harvested trees into a common database so that robust and widely applicable *AGB* models can be built.

By contrast, we found that the regional ILUA2 model developed to predict *AGB* for Zambia's forests as part of the national forest inventory severely underestimated *AGB*. There are two plausible reasons for this underestimation. Firstly, although the ILUA2 model was developed using 1319 harvested trees, these comprised mostly small to medium size trees ($D \leq$ 30cm) (Forestry Department, 2016). Small sample sizes and/or large sample sizes with mostly smaller trees to parameterise *AGB* models have been identified as major sources of *AGB* errors and uncertainty (Chave et al., 2004). Secondly, the ILUA2 model uses $D$ as the sole predictor of *AGB*. While $D$ is clearly the single strongest correlate of *AGB* (Fig. 1), in doing so, the model does not account for differences in *H-D* scaling among species (Fig. S6) and forest types (Timberlake et al., 2010; Munalula et al., 2020; Yambayamba et al., 2024), nor does it incorporate variability in wood density which can substantially affect *AGB* estimates (Mitchard et al., 2014; Jucker et al., 2018; Phillips et al., 2019). Our findings align with a recent study in the same region suggesting that commonly applied allometric models underestimate *AGB* of



miombo woodlands (Demol et al., 2024). This has important practical implications, including
reporting and verification of forest carbon stocks and dynamics under REDD+ (Köhl et al.,
2020; Grz, 2021; Melo et al., 2023) as well as emerging carbon trading initiatives. For instance,
Zambia has previously used the ILUA2 model to estimate forest reference emissions level
submissions to the United Nations Framework Convention on Climate Change (UNFCCC)
(Grz, 2016) but most recently identified the pantropical model for consideration (Grz, 2021).
In this respect, our study highlights the challenges associated with accurate estimation of $AGB$
stocks and therefore underscores the need for rigorous methods of quantifying $AGB$ stocks and
dynamics in miombo woodlands.
**4.2 Tree height uncertainty can severely bias $AGB$ estimates**
Our study reiterates the recommendation of previous works to account for tree height when
estimating $AGB$ (Feldpausch et al., 2012; Chave et al., 2014; Djomo et al., 2016; Handavu et
al., 2021). However, our results also underscore how the inclusion of $H$ in $AGB$ models is not
without challenges (Chave et al., 2014; Kapinga et al., 2018). Firstly, conventional field
measurements of $H$ are not only time consuming but also prone to errors, especially for large
trees and in closed canopies (Larjavaara and Muller-Landau, 2013; Wang et al., 2019).
Secondly, the common practice of using $H$-$D$ models to estimate tree height when this is not
measured is equally subject to errors, especially where locally fitted $H$-$D$ models are
unavailable (Fayolle et al., 2016; Sullivan et al., 2018). In line with previous studies, our results
demonstrate the peril of applying generic and regional $H$-$D$ models to specific locations that
are environmentally or taxonomically distinct (Feldpausch et al., 2012; Kearsley et al., 2013;
Molto et al., 2014; Ledo et al., 2016; Fayolle et al., 2016). Using a regional $H$-$D$ model
developed for tropical dry forests led to a severe underestimation of $AGB$. Worse still, this
underestimation increased with $D$ (Fig. 4f), which is particularly undesirable since larger trees
contribute disproportionately to total $AGB$ (Fig. 1;(Lutz et al., 2018)). Our approach of fitting





a locally-calibrated and species-specific *H-D* model substantially improved the accuracy of
*AGB* estimates, but access to high-quality local datasets to parameterise such allometries is
often limited (Jucker et al., 2022).
The importance of tree height in *AGB* estimation has also been demonstrated in studies
using remote sensing methods to map forest structure. For example, detailed reconstructions of
tree architecture made possible by terrestrial laser scanning have shown that traditional
allometric models can underestimate tree height and therefore *AGB* (Calders et al., 2022;
Terryn et al., 2024). Given the obvious challenges associated with accurate measurement of *H*
using conventional field methods, we argue that improvement in *AGB* estimation in miombo
woodlands requires leveraging increasingly available and affordable remote sensing
technologies such as LiDAR (Momo Takoudjou et al., 2018; Calders et al., 2022; Demol et al.,
2022; Demol et al., 2024). These data provide a robust way to characterise the 3D architecture
of trees that can be used to describe forest structure, functioning and dynamics with greater
accuracy and across larger scales than traditional field methods (Jucker et al., 2023; Battison
et al., 2024).
**4.3 The importance of accounting for wood density when estimating *AGB***
Wood density can vary considerably among tree species, with values ranging almost three fold
from as low as 0.34 g cm$^{-3}$ to 0.86 g cm$^{-3}$ across those sampled in our study. As previous work
in Amazonia has shown, failure to properly account for this variability in wood density among
species and forest types can severely bias large-scale estimates of forest carbon stocks inferred
via remote sensing (Mitchard et al., 2014; Phillips et al., 2019). In the case of the miombo
woodlands studied here, uncertainty in wood density was of secondary importance to that in
tree height in driving errors in *AGB* estimation. But this is likely to differ among forest types
and will depend on both the accuracy of available *H–D* allometries and the coverage of wood
density databases like the GWDD.





When field-measured values of wood density were replaced with ones obtained from
the GWDD, we found that overall wood density was overestimated across the population. This
led to a modest reduction in underestimation of $AGB$, as wood density values from the GWDD
were generally moderately correlated to those measured in the field (Fig. S7). This confirms
previous work showing that while wood density can vary somewhat within species, it is
generally quite a conserved trait (Chave et al., 2009). Therefore, provided trees have been
identified to species level and that these species have matching observations in the GWDD,
our results suggest that $AGB$ can be estimated with limited bias. This provides a strong
incentive to continue to curate and expand field reference collections of wood density to
support large-scale mapping of forest carbon stocks (Mo et al., 2024). In the context of miombo
woodlands in Zambia, only 31 tree species (out of approximately 238 uniquely identified
species in the most recent extensive national forest inventory (Pelletier et al., 2018;
Yambayamba et al., 2024)) have wood density records (Forestry Department, 2016), making it
imperative that taxonomic coverage is increased in the future.
Uncertainty in wood density can compound the errors in $AGB$ associated with the use
of a generic $H$–$D$ allometry, but it is easy to imagine a situation in which these errors could
instead cancel each other out, making them even hard to detect and quantify. In this regard, our
framework provides a simple and robust way to decompose errors in $AGB$ arising from
uncertainty in both height and wood density. Replicating this analysis on a larger dataset, such
as the pantropical $AGB$ database developed by Chave et al. (2014), would provide valuable
insights into the main sources of error in $AGB$ estimation and how these vary across forest
types and biogeographic regions.



**ACKNOWLEDGEMENTS**
AMY was supported by a PhD scholarship through the Commonwealth Scholarship
Commission (CSC) programme (grant: ZMCS-2021-542). TJ was supported by a NERC
Independent Research Fellowship (grant: NE/S01537X/1) and through a UKRI Frontier
Research grant (grant: EP/Y003810/1) that also supported AMY.
**AUTHOR CONTRIBUTIONS**
AMY and TJ conceived the idea for the study. AMY did the data collating and led the analysis
with assistance from TJ. All authors contributed substantially to revisions.
**CONFLICT OF INTEREST**
The authors declare no competing interests.
**DATA AND CODE AVAILABILITY STATEMENT**
Data and R code to replicate the results of this study will be publicly archived on Zenodo
following the review of this paper.
**SUPPLEMENTARY MATERIAL**
Additional supporting information may be found online in the Supporting Information section.



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
