# Peer review of "Tree height uncertainty biases aboveground biomass estimation more"

_EGUsphere, 2025_

## Referee Comment (RC1)

**Tree height uncertainty biases aboveground biomass estimation more than wood density in miombo woodlands**

1) Overall feedback:

Well written, relevant topic. Overall, the paper is carefully presented. The method section requires more detail imo, for the manuscript to be more deeply reviewed. Right now, because of a lack of information on some accounts it was not possible to correctly review it imo (therefore I also did not read the discussion in too great detail at this stage). I suggest a major revision.

-Authors should provide recommendations for how to improve agb measurements given the practical implementation and scope of their article, and this agb measurement methodology (e.g. forest carbon projects).

-Authors should clarify and potentially rephrase/concretize their objectives (L103-107)and then link it better to the methodology and data collection that was necessary for reaching their objectives.

-Authors should consider whether the introduction and article in general should speak about a larger context of tropical forests (or even 'and savannas' as they mention in first line of intro), or specifically target miombo woodlands as the title suggests. Though I understand that tropical forests in general might be relevant since equations are used in miombo from tropical forests more widely, it is important to clarify in the text the distinction imo, and also clearly introduce miombo as a specific type of dry forest at some point in the intro already. I also think the authors should acknowledge and provide info on the available other destructive tree datasets / equations for miombo systems. Please see my specific feedback on that.

-Authors should provide more information on how they tackled the fact that their data comes from three different sites (statistically / ecologically)?

-Authors should provide more detailed information on the methodology used for the forest inventory data. See specific comments.

-Authors should explain why other allometric equations from miombo (destructive tree harvest datasets) have not been included in the comparison and/or include them throughout the analysist o make it more robust and exhaustive.

2) Specific feedback:

ABSTRACT

L30 What do you mean with 'generic information'? Please clarify in the abstract.

L30 Not sure that 'in particular' fits well here as a wording, since you haven't given anything else before that in terms of error demonstration?

L32 Is 'intuitive' the correct word here? (defined as 'using or based on **what one feels to be true** even without conscious reasoning; instinctive.')

The final sentence of the abstract reads odd to me. What is the takehome message from your results? What should people calculating biomass in miombo do now based on what you found? Not all of us will be able to partition errors if we do not have destructive tree data so do you recommend a middle way solution to limit the errors but still have the potential to monitor without cutting trees… Is the recommendation about using specific height measurement methods, or measuring

height more repetitively to lower bias? Is it about needing more wood density values from actual trees in the field?

Shouldn't the keywords be new words compared to the title and not repetitive?

INTRO

L47-52 Though I do not think addressing my comment would need to be done exactly here in the article, I think the authors should also address the uncertainty/bias in carbon projects where numbers from ecosystem- or site-specific contexts are transferred to a completely different context, out of the assumption that variation in the biomass amounts will be negligible. Large variation in small-scale environmental conditions as well as land-use history and ongoing management (eg. fire) in miombo eg might mean the biomass values can differ a lot between sites and regions. I could not find a review that has addressed this already (based on field data and not remote sensing only), but I think it requires touching upon this variation in the current paper at least as well. I think the sentence 'this is rarely the case even at the most basic level of the individual tree' made a nice jump for me to think about this, since you could then extend this to say 'even on the level of the site or region there is much room for bias/less room for simple comparisons'. Also with 'there are multiple sources of uncertainty that affect tree biomass estimation' → You could present here some of the 'other uncertainty' except for the ones you look at in your analyses (which are methodological), cfr. environmental, management variation etc.

L62 Should the authors touch upon some more recent efforts here and innovations towards the future, eg. ESA 2025 Biomass launch?

L65 I think there needs to be more depth provided here. What are the myriad of equations? Can you provide a table with full overview? Are you referring only to equations for tropical forests (cfr 2nd part of your sentence), or all biomes? How do you demonstrate that the most widely used is the ones by Chave? Do you mean they are most widely used in science or in forest carbon monitoring & reporting, where other equations might be used than the published ones but this information might not be publically available (?), despite it being used to monitor carbon stocks as well across many different areas.

Related to this comment, I miss information on which previous datasets exist that target similar research questions for miombo specifically (destructive tree harvesting) though maybe that will come later on in the article. Eg referring to some of these studies (probably non-exhaustive).

https://www.sciencedirect.com/science/article/pii/S0167880912001892

https://www.sciencedirect.com/science/article/pii/S0378112713005306

https://www.sciencedirect.com/science/article/pii/S0378112712007074

https://onlinelibrary.wiley.com/doi/full/10.1111/j.1744-7429.2010.00713.x

https://www.sciencedirect.com/science/article/pii/S0378112717308071 -- including one of the authors to this current article, so please clarify how this dataset links to the current article's dataset?

L70 Please explain 'measured inputs', I did not understand this contradiction with the line before.

L69-74 On several occasions, I miss scientific backing. Eg here, there is a lack of references to support your statements. Please introduce these.

L86 Interesting, I wonder if the authors can give concrete examples for the availability of wood density for miombo ecosystems, since their article targets this biome. How much % of the species wood density is available for this biome? Have there been efforts to sample wood density on a large scale?

The introduction in general is very broad on ecosystems the authors seem to cover with their article (tropical forests and savannas in first line of intro), while the article is supposed to talk about miombo. I think this needs clarification as to whether the authors will stick in the introduction to broad context, tropical forests, or whether miombo should be more specifically mentioned from the start.

L94 Can the authors touch upon what previous work has been done in the same context? Have there been other quantitative assessments like this and what have they found?

L98 I'm not sure if comparing miombo area to the size of the UK is super helpful in a global context. Could the authors find a better comparison related more to the study area/biome's location?

L104 Please clarify 'new biomass allometric models fit to the data' to make it more clear. Please explain or introduce earlier the availability of 'local biomass models' cfr my previous comment.

L106-107 It would be useful at this stage to get more information on what types of approaches you tested on imputing H and wood density. Is it about different measurement methods/protocols, or only different computer based methods?

METHODS

2.1

L110 'Multiple datasets' – does it cover multiple study areas/regions? It should be clarified, which comes later in 2.2 and Table 1 but ideally is already mentioned here more concretely and in the introduction when the authors introduce 154 trees were used. It seems important information that there is also a site-level variable that may need to (have been) addressed in the statistical analyses. Cfr also my comment on 'other uncertainty drivers' where I touch upon site/region level variation in agb estimates.

L111 Please provide a reference

L114 A map would be helpful, for the reader audience that is not familiar with miombo and their region of occurrence

L115 Can you make this concrete, 'long dry seasons', 'months of intense rainfall', 'fire playing a key role'? Can you touch upon the difference between dry and wet miombo as well?

L117 Can you provide references for the statement 'trees in this region … are able to recover remarkably quickly from wildfires and other forms of disturbance…'?

2.2

L132 Can you provide argumentation why the outliers were excluded?

Table 1: it clearly shows you have looked at the site level variation because you include elevation, MAP and MAT – where in the article do you touch upon how you have addressed the fact that combined dataset comes from 3 different sites and how you accounted for this potential influential factor in your analyses and results?

2.3

Can the authors provide more information on the measurement methodology of the forest inventory data? As well as the spatial locations of the destructive harvesting vs. the inventory data? This seems like crucial information to understand and assess how robust the comparative analyses in that is the key of the article (how does height and wood density measurements influence the biomass quantification with field data vs harvesting data). If this data has been published, the authors can refer to the specific publication for detailed methodology information.

L146 What was the reasoning for excluding the outliers?

There is a section explaining H-D dataset, but what about what was done for wood density? Where can we find information on it? Also, for Height, what exactly has been compared in terms of testing the influence of H imputation on biomass quantification?

2.4

Would be helpful to know what the different packages are used for, as now it is just a summation of many different packages which gives limited information.

Overall this section seems invaluable for me right now. It is not clear what was done as part of data harmonization and analyses in relation to the two research objectives*. At this stage, it became clear for me that the authors should clarify and potentially rephrase/concretize their objectives and then link it better to the methodology and data collection that was necessary for reaching their objectives.

*L103-107

*Using these data, we set out to address **two key research objectives**. First, we compared AGB estimates obtained using new biomass allometric models fit to the data with those of existing local and pantropical biomass models. Second, we systematically assessed how different approaches to imputing H and $\rho$ affect tree-level AGB estimates.*

2.5

The authors should work towards improving this section to be very clear and linked to their research objectives. Please specify here already what it means 'newly developed biomass models', despite explaining it later on in this section.

L164 Can the authors reflect upon the other available allometric equations from Miombo woodlands (cfr previous comment)? Why was the Ilua 2016 equation included here as comparison and not the other available allometric equations?

Can the authors also reflect upon the following article https://www.mdpi.com/1999-4907/7/2/13 ?

L170 Why these two options were selected, combined rho diameter² height vs. diameter alone? Please explain. It is unclear to me why in figure 1 there is then also a correlation shown with rho or height alone? Please also explain how the other available miombo allometric equations fit into your storyline and why they were not mentioned or compared as well.

L181 Please provide a backing for your statement (larger trees, largest errors). Please specify 'absolute terms of what'.

Figure 1: can the authors also give information on the tree inventory dataset? I'm not sure if figure 1 is not better fit for incorporation into the results since you are including this tree harvest dataset model as a part of your comparison or research objective 1? (L167-171). Can the authors argue why it should be in methods and is not a result?

2.6

L216 Shouldn't this decision of which model to use for the height and wood density influence be based on what your previous analyses showed as the 'best model' for your dataset, rather than the most commonly used model?

L229 Which 4321 trees?

L231 I think the authors should state clearer throughout the paper how the uncertainty from height/wood density was assessed, by referring to the Bayesian technique at some more occasions. This aspect remained very unclear for me until this point in the article, and it would help readers to clarify that more from the start. Also the separation and methods of the two distinct research objectives and why the authors focus on both should be better clarified I think.

I believe I understand it like this but it took me a while to figure it out and the authors could help the readers to clarify that easier: First there's the comparison of 2 existing (based on destructive tropical forest as well as miombo woodland harvest data) and 2 new models (based on their own destructive miombo data) to see how the regression models perform in terms of predicted vs observed (field inventory data) and when including combinations of predictors vs single predictors. Then there's the check of the field based height and wood density are influencing the uncertainty in the model based on field inventory data and Bayesian modelling, using only the chave model.

RESULTS

It is problematic for me to see that the site variation has not been tackled or addressed in this analysis. The destructive data comes from three different sites – where is the argumentation from the authors that state/show that they can be used as a combined dataset?

3.1

L264-265 Please explain in caption the three numbers what they refer to each time (RMSE 160.5 vs 210.6 vs 217.9 eg.). Why are you only showing it for 3 of the 4 models?

Figure 2

      Can the authors specify the model formula in a and b as well like they do for c and d?

      Can they explain in caption what a, b stands for?

      Can e be shown in the same order as they show the models (ILUA2, Chave, Fitted1, Fitted2)?

L266 Please explain how the ILUA2 model was developed more clearly in the methods where introducing this model. If not, it would at least be expected to be in the discussion, considering the underestimation.

3.2

L297 More information is needed on the field inventory dataset and measurement protocol to understand what height and rho values were measured (how many replicates, which instrument etc)?

L302 I think you should rephrase it so it's clear that it's the difference (higher values) that leads to the possible reduction in underestimation, not the fact that they're correlated.

It would be good throughout this section to reiterate what species-specific vs regional level estimates meant in terms of method. That's really the key difference in the paragraphs here so should be 100% clear for readers.

Some of this end section in results already read more like discussion (compounded errors etc).

Figure 3

Can you make it more visual clear the different scenarios? Eg by showing on the left = measured H all three panels, then middle is Species H, right is Regional H (and similar for top to bottom the differences)? Maybe also by using color codes or size for variation in the RMSE and PE (so we can easily see which ones have smaller or larger errors?)

It isn't mentioned that Bayesian modelling was involved – could this be mentioned in the figure caption of those figures that involved it (?figure 4).

L307-320 This section involved 3 figures as output and the information is not separated clearly enough for readers to digest it. Please try to make this clearer what you found in every step and every figure.

The abstract should become more clear and concrete when having read the results now. You mention in L30 'using generic information' – please make this specific and linked to how you name the 9 scenarios in the actual article, eg measured, species, regional, …

DISCUSSION

L385-387 Please explain more carefully why this priority stems from the previous lines of text

I miss a section where the authors zoom out on their results and bring forward 'lessons learnt' for people measuring and utilizing numbers on carbon stocks in practice. Generally, the discussion ends on a very specific statement and no any conclusion is made.

---

## Referee Comment (RC2)

**Tree height uncertainty biases aboveground biomass estimation more than wood density in miombo woodlands**

**Review – Miro Demol**

1. **Overall Feedback**
   a. **What I liked**

The Authors present a clear and concise study decomposing the effects on AGB estimations of wood density and tree height data source alternatives when direct measurements are not available. The study uses data from two destructive validation experiments (previously published) in miombo woodlands and applies a Bayesian framework for parametrizing models. The results are readily applicable for carbon quantification of miombo woodlands, for instance for national accounting or REDD+ initiatives. The text is well written and clear, with a good selection of graphs (a few minor specific comments below).

   b. **What should be improved**

Citations in Introductions – For some statements, citations seem superfluous, for others they are lacking. E.g. do you need 4-5 citations for what now is well-established (L40, L43, …)? L88 Réjou-Mechain is I think not the right citation to claim 'almost always assigned from databases' – as it is a methodological paper introducing R Biomass package.  L64, do you really need the two Chave citations (which are repeated on L66 again)? But it would be nice to get citations for e.g. L69 (pantropical allometries performing very well on destructive data) or L85 (an example of H-D models overlooking forest type dependence).

Incorporation of other miombo destructive data – The destructive data used here (Handavu et al. 2021, Kapinga et al 2018) is valuable bit fairly limited e.g. in D range (max value of 52 cm). The limited geographical range of the calibration data can cause overfitting and spatial autocorrelation. The leave-one-out cross-validation will help with the first issue, but not with the second. I suggest incorporating more destructive data from miombo woodlands (for instance the Mugasha et al. (2013) dataset that spans 150+ trees ranging up to 110 cm D). Since no data was collected for this study as far as I can tell (the data comes from previously published papers) I would have expected a broader search for reference data in literature.

Circular reasoning in local H-D - The "locally-calibrated species-specific allometry" (Eq. 7) is fit using data from the same sites where the 154 harvest trees come from, and then predicted heights are used in Chave 2014 to estimate AGB for those same 154 trees. The scenario in Fig 3b shows "no bias" but this is partly because the H-D model was calibrated on overlapping/same data. This should be acknowledged as a limitation - the "species-specific allometry" scenario is not truly independent.

2. **Specific Feedback**

| Line | Comment |
|------|---------|

| L76 | What's causing this bias? Is H usually over or underestimated? |
|---|---|
| L 79 | Should be D instead of H – as in a bias in input D to infer H? Or do you mean underlying bias in the calibration H data to construct H-D models? |
| L98 | It would be good to quantify 'substantial' |
| L163 | Please explain the bias correction factor in the equation |
| L172-177 | I don't think RMSE and bias need a citation as they're well-established model performance metrics. Replacing L172-175 with "We then compared the predictive ability of these four models with the root mean square error (RMSE) and percentage bias (PE):" would be clear enough. |
| L187 | I've got difficulties interpreting Fig 1a. Why is the red 14% bar larger than the 29% bar? |
| L240 | Maybe a figure or table with the 3x3 matrix with the nine scenarios will help to understand these permutations more quickly? |
| L240 | Consider change 'population' to data set mean |
| L320 | This figure could be improved in my opinion. Again as a 3x3 matrix, with shared axis to make maximal use of the space, with on the top the H gradient and to the right the density gradient as titles. Panel (a) is repeated from Fig 2, so I would omit it in Fig 2. |
| L322 | I wonder if coarser attribution level of WD made predictions worse. |
| L367 | There is an artefact in the height data – values center around integer values – or is this intentional? What were the methods to measure tree height in the 4321 trees data? |
| L388 | Apart from tree size, is the forest structure, species composition, etc. sufficiently similar between the ILUA2 calibration data set to apply the ILUA2 model on the data in this study? |
| - | Would it make sense to swap the discussion of the WD and H effects with the Chave vs ILUA2? I.e. get to the point you're making in the title of the paper first? |

---

## Author Comment (AC1)

**Referee 1**

**Overall feedback**

**Comment**: Well written, relevant topic. Overall, the paper is carefully presented. The method section requires more detail imo, for the manuscript to be more deeply reviewed. Right now, because of a lack of information on some accounts it was not possible to correctly review it imo (therefore I also did not read the discussion in too great detail at this stage). I suggest a major revision.

**Response:** Thank you very much for taking the time to review our paper and for providing valuable feedback on how to improve it. We are pleased to hear that overall you enjoyed it. See below for a point-by-point response to your suggestions on how to improve the clarity of the methods.

**Comment:** Authors should provide recommendations for how to improve agb measurements given the practical implementation and scope of their article, and this agb measurement methodology (e.g. forest carbon projects).

**Response**: This is a good suggestions and we intend to incorporate this element in the revised discussion of the paper.

**Comment:** Authors should clarify and potentially rephrase/concretize their objectives (L103-107) and then link it better to the methodology and data collection that was necessary for reaching their objectives.

**Response:** We agree that it is important to ensure that the objectives and methodology are well aligned. See below for specific responses to how we intend to improve this element of the paper when revising it.

**Comment:** Authors should consider whether the introduction and article in general should speak about a larger context of tropical forests (or even 'and savannas' as they mention in first line of intro), or specifically target miombo woodlands as the title suggests. Though I understand that tropical forests in general might be relevant since equations are used in miombo from tropical forests more widely, it is important to clarify in the text the distinction imo, and also clearly introduce miombo as a specific type of dry forest at some point in the intro already. I also think the authors should acknowledge and provide info on the available other destructive tree datasets / equations for miombo systems. Please see my specific feedback on that.

**Response**: Thank you for this feedback. While our analysis leverages data from miombo woodlands, the overarching aim of our study is to provide a general framework for assessing the sources of uncertainty in tree AGB estimation that could be applied to any ecosystem. This is why we chose to keep the focus of the introduction broader than just the miombo. We do agree however that in the final paragraph of the introduction we could make a stronger case for why miombo woodlands provide a good testbed for our analysis. In revising the introduction of our paper, we therefore plan to provide a better rationale for why testing our framework in the miombo is relevant and timely.

**Comment:** Authors should provide more information on how they tackled the fact that their data comes from three different sites (statistically / ecologically)?

**Response**: As we explain in more detail below, the three study sites from which we compiled data are actually bioclimatically very similar within the context of miombo woodlands. We will explain this more clearly in the revised paper and will also add a figure to the appendix to support this claim by showing that the AGB allometries of trees from these sites are indistinguishable.

**Comment**: Authors should provide more detailed information on the methodology used for the forest inventory data. See specific comments.

**Response**: Thank you for this useful feedback. See below for details on how we plan to address these specific points.

**Comment:** Authors should explain why other allometric equations from miombo (destructive tree harvest datasets) have not been included in the comparison and/or include them throughout the analysist to make it more robust and exhaustive.

**Response**: The reason for this is that other destructive harvest data from the miombo lack direct measurements of wood density. As a result, these data cannot be used within our methodological framework. We will clarify this key requirement for including data in our analysis in revising the methods.

**Specific feedback**

**Abstract**

Comment: L30 What do you mean with 'generic information'? Please clarify in the abstract.

**Response:** We agree that this should have been clearer. Generic information here implies using height-diameter models and/or wood density values that are not specifically derived from miombo woodlands data sources. To make this point clearer we plan to rephrase this as: '... are imputed using data that are not biome specific'.

**Comment**: L30 Not sure that 'in particular' fits well here as a wording, since you haven't given anything else before that in terms of error demonstration?

**Response:** 'In particular' was used to refer to the generic height-diameter model (for tropical dry forests) used to estimate height. Having now revised the previous sentence as suggested this should be clearer, but also plan to replace in particular with 'Specifically' to avoid confusion.

**Comment**: L32 Is 'intuitive' the correct word here? (defined as 'using or based on what one feels to be true even without conscious reasoning; instinctive.')

**Response**: Agreed, we plan to remove this as it is superfluous.

**Comment**: The final sentence of the abstract reads odd to me. What is the take home message from your results? What should people calculating biomass in miombo do now based on what you found? Not all of us will be able to partition errors if we do not have destructive tree data so do you recommend a middle way solution to limit the errors but still have the potential to monitor without cutting trees... Is the recommendation about using specific height measurement methods, or measuring height more repetitively to lower bias? Is it about needing more wood density values from actual trees in the field?

Response: Thank you for this observation. Our goal with this final sentence was to highlight that a similar exercise to ours could be conducted using other available datasets where destructive harvests have been performed. This would help identify the relative contribution of wood density and height as drivers of uncertainty in AGB across a range of ecosystem types. We agree however, that what is missing from our abstract is a specific recommendation relating to the miombo. To this end, we plan to add a sentence before this final statement in which we advocate for the need for better height-diameter data and models from the miombo. We also plan to revise the final statement of the abstract as follows to make it clearer that our approach could be expanded to other woody ecosystem types: '... paving the way for more robust estimation of forest carbon stocks and their uncertainties across a range of ecosystem types'.

Comment: Shouldn't the keywords be new words compared to the title and not repetitive?

**Response**: Thank you for noting this, we will revise the key words to exclude ones that are in the title.

**Introduction**

Comment: L47-52 Though I do not think addressing my comment would need to be done exactly here in the article, I think the authors should also address the uncertainty/bias in carbon projects where numbers from ecosystem- or site-specific contexts are transferred to a completely different context, out of the assumption that variation in the biomass amounts will be negligible. Large variation in small-scale environmental conditions as well as land-use history and ongoing management (eg. fire) in miombo eg might mean the biomass values can differ a lot between sites and regions. I could not find a review that has addressed this already (based on field data and not remote sensing only), but I think it requires touching upon this variation in the current paper at least as well. I think the sentence 'this is rarely the case even at the most basic level of the individual tree' made a nice jump for me to think about this, since you could then extend this to say 'even on the level of the site or region there is much room for bias/less room for simple comparisons'. Also with 'there are multiple sources of uncertainty that affect tree biomass estimation' 'You could present here some of the 'other uncertainty' except for the ones you look at in your analyses (which are methodological), cfr. environmental, management variation etc.

**Response:** We agree with this suggestion and plan to expand this argument beyond the level of the individual tree when revising our paper. Specifically, we intend to discuss how the use of biomass estimates derived from one ecosystem type or environmental context and applied to another can lead to similar issues with bias and uncertainty at the stand and landscape scale.

**Comment**: L62 Should the authors touch upon some more recent efforts here and innovations towards the future, eg. ESA 2025 Biomass launch?

**Response**: Thank you for the suggestion, we will mention these recent developments in biomass mapping when revising the introduction.

**Comment**: L65 I think there needs to be more depth provided here. What are the myriad of equations? Can you provide a table with full overview? Are you referring only to equations for tropical forests (cfr 2nd part of your sentence), or all biomes? How do you demonstrate that the most widely used is the ones by Chave? Do you mean they are most widely used in science or in forest carbon monitoring & reporting, where other equations might be used than the published ones but this information might not be publically available (?), despite it being used to monitor carbon stocks as well across many different areas.

Response: We will clarify that we mean that there are numerous (hundreds if not thousands) of allometric models that have been developed for estimating tree biomass and wood volume across different forest types. To support this, we will cite relevant literature that has made attempts to summarise and compile these models (e.g., FAO's GlobAllomeTree database, Zianis et al. 2005 for Europe and Chojnacky et al. 2014 for North America). As far as our assertion that the models developed by Chave are widely used in the tropics, this is supported by the fact that these equations have been used in almost 10,000 published articles. They are also the default used in many biomass estimation routines and workflows, such as the CEOS protocol and the BIOMASS package. Nevertheless, to avoid confusion, we will rephrase the sentence to say that Chave's models are 'among the most widely used for tropical trees' (rather that the most).

**Comment:** Related to this comment, I miss information on which previous datasets exist that target similar research questions for miombo specifically (destructive tree harvesting) though maybe that will come later on in the article. Eg referring to some of these studies (probably non-exhaustive).

- https://www.sciencedirect.com/science/article/pii/S0167880912001892
- https://www.sciencedirect.com/science/article/pii/S0378112713005306
- https://www.sciencedirect.com/science/article/pii/S0378112712007074
- https://onlinelibrary.wiley.com/doi/full/10.1111/j.1744-7429.2010.00713.x
- <a href="https://www.sciencedirect.com/science/article/pii/S0378112717308071">https://www.sciencedirect.com/science/article/pii/S0378112717308071</a> -- including one of the authors to this current article, so please clarify how this dataset links to the current article's dataset?

**Response:** Thank you for providing these references. We agree that it is important to provide some more context on previous research in the miombo that has developed AGB equations based on destructive harvesting. We intend to do this in the 'Study system' section of the methods.

**Comment**: L70 Please explain 'measured inputs', I did not understand this contradiction with the line before.

**Response**: With 'measured inputs' we referred to height and wood density. We will clarify this when revising our paper.

**Comment**: L69-74 On several occasions, I miss scientific backing. Eg here, there is a lack of references to support your statements. Please introduce these.

**Response**: Thank you for pointing this out, we will make sure to back up these statements with appropriate references. In this specific case, we had not included the references as they are cited in the preceding paragraph where we introduce these AGB allometric models developed by Chave et al. We will correct this in revising our paper.

**Comment:** L86 Interesting, I wonder if the authors can give concrete examples for the availability of wood density for miombo ecosystems, since their article targets this biome. How much % of the species wood density is available for this biome? Have there been efforts to sample wood density on a large scale?

**Response**: Good suggestion, we will provide information on the percentage of tree species with wood density values in the miombo woodlands.

**Comment**: The introduction in general is very broad on ecosystems the authors seem to cover with their article (tropical forests and savannas in first line of intro), while the article is supposed to talk about miombo. I think this needs clarification as to whether the authors will stick in the introduction to broad context, tropical forests, or whether miombo should be more specifically mentioned from the start.

Response: While our analysis leverages data from miombo woodlands, the overarching aim of our study is to provide a general framework for assessing the sources of uncertainty in tree AGB estimation that could be applied to any ecosystem. This is why we chose to keep the focus of the introduction broader than just the miombo. We do agree however that in the final paragraph of the introduction we could make a stronger case for why miombo woodlands provide a good testbed for our analysis. In revising the introduction of our paper, we plan to provide a better rationale for why testing our framework in the miombo is relevant and timely. Specifically, miombo woodlands cover around 2 million km² of central and southern Africa (an area approximately the size of Spain, France, Germany and Italy combined) and store surprisingly large amounts of biomass in their woody vegetation. However, they remain severely understudied relative to other ecosystems, and when it comes to the estimation of the AGB it is not uncommon for researchers to rely on allometric equations developed for other tropical forest regions despite clear differences in their structure, species composition and climate. In this respect our paper aims to develop a general approach to deconstructing sources of uncertainty that can be applied in an ecosystem where need for better AGB allometries is high.

**Comment**: L94 Can the authors touch upon what previous work has been done in the same context? Have there been other quantitative assessments like this and what have they found?

Response: While there have been studies that have explored the influence of tree height and wood density on AGB estimates in isolation – which we will cite here explicitly when revising our paper – to the best of our knowledge ours is the first to systematically deconstruct their combined contributions to AGB errors. Nevertheless, the paper by Molto et al 2012 (<a href="https://doi.org/10.1111/j.2041-210x.2012.00266.x">https://doi.org/10.1111/j.2041-210x.2012.00266.x</a>) explains something somewhat very similar to us, although not using the exact same framework and found that tree height and wood density contributed negligible uncertainty to AGB estimates in a tropical forest in French Guiana.

**Comment**: L98 I'm not sure if comparing miombo area to the size of the UK is super helpful in a global context. Could the authors find a better comparison related more to the study area/biome's location?

**Response**: We agree and will revise this sentence by explicitly stating the extent of the miombo biome and clarifying that it is the largest dry forest and savanna ecosystem of its type anywhere in the world.

**Comment:** L104 Please clarify 'new biomass allometric models fit to the data' to make it more clear. Please explain or introduce earlier the availability of 'local biomass models' cfr my previous comment.

**Response**: Agreed, we will clarify here that the data were used to derive a new AGB allometry.

**Comment:** L106-107 It would be useful at this stage to get more information on what types of approaches you tested on imputing H and wood density. Is it about different measurement methods/protocols, or only different computer based methods?

Response: We feel this level of detail is best left for the methods rather than the brief summary at the end of the introduction. To avoid making this section too detailed and repeating ourselves in the methods, we would propose to keep this section as is.

**Methods**

**Comment**: L110 'Multiple datasets' – does it cover multiple study areas/regions? It should be clarified, which comes later in 2.2 and Table 1 but ideally is already mentioned here more concretely and in the introduction when the authors introduce 154 trees were used. It seems important information that there is also a site-level variable that may need to (have been) addressed in the statistical analyses. Cfr also my comment on 'other uncertainty drivers' where I touch upon site/region level variation in agb estimates.

**Response**: Agreed. We will rephrase this sentence to simply say that 'The study was conducted using tree harvest data acquired in Zambia's miombo woodlands', removing the mention of multiple datasets which are then described below. Regarding bioclimatic differences among the three study sites, see responses to specific points below.

**Comment**: L111 Please provide a reference

**Response**: We will provide a reference describing the overall biogeography and ecology of the miombo.

**Comment**: L114 A map would be helpful, for the reader audience that is not familiar with miombo and their region of occurrence

**Response:** We will provide a map with the locations of the three sites in the appendix and refer to it here.

**Comment**: L115 Can you make this concrete, 'long dry seasons', 'months of intense rainfall', 'fire playing a key role'? Can you touch upon the difference between dry and wet miombo as well?

**Response**: We will add specific details about the duration and timing of the dry season, the annual rainfall and how these vary geographically across the miombo region. We will also provide a reference to support the statement on fires being a major agent of disturbance in the region.

**Comment**: L117 Can you provide references for the statement 'trees in this region ... are able to recover remarkably quickly from wildfires and other forms of disturbance...'?

**Response**: Previous work has shown that stand-level AGB in miombo woodlands recovers to pre-disturbance levels in as little as 25-40 years (Williams et al. FEM 2008; McNicol et al. EcolApp 2015). We will add these references to the revised paper.

Comment: L132 Can you provide argumentation why the outliers were excluded?

**Response**: The outliers were excluded based on visual inspection of scatter plots, as shows in the figure we provide in the appendix. We will clarify this in the methods.

**Comment:** Table 1: it clearly shows you have looked at the site level variation because you include elevation, MAP and MAT – where in the article do you touch upon how you have addressed the fact that combined dataset comes from 3 different sites and how you accounted for this potential influential factor in your analyses and results?

Response: Thank you for raising this important point. All three sites come from woodlands classified as 'wet miombo' based on their mean annual rainfall, with the cut-off generally considered to be at 1000 mm/yr. In this regard all three of our sites can be considered bioclimatically very similar within the wider context of the miombo, which is the reason we chose to focus on them and felt confident in combing their data. To illustrate that the AGB allometries of trees from these sites are not distinguishable, we will add a figure to the appendix with colour-coded points for each site and refer to it in this section of the methods. It is worth noting that the climatic variation between our sites is small not only in the context of the miombo, but also more generally when considering previous work by Chave et al. that has combined tree harvest data from across the wet and dry tropics. As our results show, even these pantropical models predict AGB with little bias when supplied with accurate estimates of height and wood density.

**Comment**: section 2.3. Can the authors provide more information on the measurement methodology of the forest inventory data? As well as the spatial locations of the destructive harvesting vs. the inventory data? This seems like crucial information to understand and assess how robust the comparative analyses in that is the key of the article (how does height and wood density measurements influence the biomass quantification with field data vs harvesting data). If this data has been published, the authors can refer to the specific publication for detailed methodology information.

**Response**: The measurement methodology of the forest inventory data is provided in the two references cited in section 2.3. We will clarify this when revising the paper and will also add the locations of sites from which the inventory data were compiled to the map of the miombo woodlands showing the location of the harvest data.

**Comment:** L146 What was the reasoning for excluding the outliers?

**Response**: The outliers were excluded based on visual inspection of scatter plots, which revealed substantial deviations from expected values which are consistent with either data entry errors or trees with broken crown stems.

**Comment:** There is a section explaining H-D dataset, but what about what was done for wood density? Where can we find information on it? Also, for Height, what exactly has been compared in terms of testing the influence of H imputation on biomass quantification?

**Response**: The information on wood density is provided in the previous section describing the tree harvest data. As for the approach we used to determine the contribution of height uncertainty on AGB errors, this is described below in section 2.6.

**Comment**: section 2.4. Would be helpful to know what the different packages are used for, as now it is just a summation of many different packages which gives limited information.

**Response**: Agreed. In addition to listing them all, we will also provide specific references in the text to how they were used to format and analyse the data.

**Comment:** Overall this section seems invaluable for me right now. It is not clear what was done as part of data harmonization and analyses in relation to the two research objectives. At this stage, it became clear for me that the authors should clarify and potentially rephrase/concretize their objectives and then link it better to the methodology and data collection that was necessary for reaching their objectives.

**Response:** We agree that the research objectives highlighted at the end of the introduction could be expanded on to make them clearer and tie them more directly to the methodology described later on. In revising the paper we plan to expand this section of the introduction accordingly.

**Comment:** L103-107. Using these data, we set out to address two key research objectives. First, we compared AGB estimates obtained using new biomass allometric models fit to the data with those of existing local and pantropical biomass models. Second, we systematically assessed how different approaches to imputing H and affect tree-level AGB estimates.

**Response**: As mentioned above, we plan to rephrase and expand on these objectives when revising the paper to ensure that they are clearer and better align with the subsequent description of the methods.

**Comment**: section 2.5. The authors should work towards improving this section to be very clear and linked to their research objectives. Please specify here already what it means 'newly developed biomass models', despite explaining it later on in this section.

**Response**: Thank you for this. We will ensure to clarify here that by 'newly developed biomass models' we mean a reparameterization of existing equations using our data.

**Comment**: L164 Can the authors reflect upon the other available allometric equations from Miombo woodlands (cfr previous comment)? Why was the Ilua 2016 equation included here as comparison and not the other available allometric equations?

**Response**: We did consider alternative models but settled on these two because: (1) the model by Chavel et al. 2014 is widely used in the tropical biomass literature (including for miombo woodlands, as supported by the references we provide); and (2) the ILUA model is the one currently being used to generate country-level AGB estimates for official reporting based on data from Zambia's NFI programme, including to calculate forest carbon budgets submitted to UNFCCC (see references provided for details on this). As such, these two alternative models represent those that are most widely used in the context of the miombo, something which we will aim to clarify further when revising our paper.

**Comment:** Can the authors also reflect upon the following article <a href="https://www.mdpi.com/1999-4907/7/2/13">https://www.mdpi.com/1999-4907/7/2/13</a>?

**Response**: Thank you for the reference. We will incorporate this citation in our paper, but note that in this analysis, the authors did not measure wood density for the harvested trees, meaning that these data would not be suitable for the approach we develop in our paper.

**Comment**: L170 Why these two options were selected, combined rho diameter2 height vs. diameter alone? Please explain. It is unclear to me why in figure 1 there is then also a correlation

shown with rho or height alone? Please also explain how the other available miombo allometric equations fit into your storyline and why they were not mentioned or compared as well.

**Response:** The reason these two option were selected is because they (1) are those used in the Chave et al. 2014 and ILUA models we were testing, and (2) because more generally they represent the most common forms used to model AGB (see Chave et al. 2005 and 2014 for a discussion on this). Regarding Fig. 1, the reason we show these correlation plots with height and wood density is simply to give the reader a better sense of the distribution of the data.

**Comment**: L181 Please provide a backing for your statement (larger trees, largest errors). Please specify 'absolute terms of what'.

**Response**: This is a simple reflection of the fact that in absolute terms (i.e., when errors are computed in units of kg rather than as a %), larger/heavier trees will inevitably be the ones with the greater uncertainty/errors. A 10% error in AGB for a small tree will be dwarfed by the same relative error observed in a large tree when calculated in terms of kg of AGB. Moreover, because large trees contribute disproportionately to stand-level AGB (see Lutz et al., 2018 GEB as a classic example), understanding and minimising errors in their AGB estimates is particularly important. We will clarify this point in revising the paper and support it with the above reference.

**Comment:** Figure 1: can the authors also give information on the tree inventory dataset? I'm not sure if figure 1 is not better fit for incorporation into the results since you are including this tree harvest dataset model as a part of your comparison or research objective 1? (L167-171). Can the authors argue why it should be in methods and is not a result?

**Response:** Information on the tree inventory dataset is provided in a separate figure (Fig. 5). The reason we feel that it is important to keep Fig. 1 in the methods is that it provides an overview of the tree harvest data (which is central to our analysis) and help the reader better understand the structure of the AGB models we are fitting and which are also described in the methods.

**Comment:** L216 Shouldn't this decision of which model to use for the height and wood density influence be based on what your previous analyses showed as the 'best model' for your dataset, rather than the most commonly used model?

**Response**: As we explain in the paper, we opted to use the pantropical model since its performance was very similar to our best-fitting model, which however has not yet been validated on independent data. We feel this choice makes the results of our analysis much more transferable to other studies that rely on Chave et al.'s equations to estimate AGB.

Comment: L229 Which 4321 trees?

**Response**: These are described in section 2.3 and Supplementary section S2 and Table S3.

**Comment:** L231 I think the authors should state clearer throughout the paper how the uncertainty from height/wood density was assessed, by referring to the Bayesian technique at some more occasions. This aspect remained very unclear for me until this point in the article, and it would help readers to clarify that more from the start. Also the separation and methods of the two distinct research objectives and why the authors focus on both should be better clarified I think.

**Response**: As mentioned in response to previous comments, in revising our paper, we intend to make our approach based on different scenarios of height and wood density uncertainty

clearer from the outset by referring to it more directly in both the abstract and introduction. In doing so, we will also clarify why it is important that to do so we first need to compare the performance of different AGB models, as this will be the natural starting point for any analysis aiming to estimate AGB with an existing allometric model.

**Comment:** I believe I understand it like this but it took me a while to figure it out and the authors could help the readers to clarify that easier: First there's the comparison of 2 existing (based on destructive tropical forest as well as miombo woodland harvest data) and 2 new models (based on their own destructive miombo data) to see how the regression models perform in terms of predicted vs observed (field inventory data) and when including combinations of predictors vs single predictors. Then there's the check of the field based height and wood density are influencing the uncertainty in the model based on field inventory data and Bayesian modelling, using only the chave model.

**Response**: Correct, this is a good summary of our approach and how it relates to our objectives. As mentioned previously, we will aim to clarify this aspect in revising our paper.

**Results**

**Comment:** It is problematic for me to see that the site variation has not been tackled or addressed in this analysis. The destructive data comes from three different sites – where is the argumentation from the authors that state/show that they can be used as a combined dataset?

**Response**: See previous responses to this point.

**Comment**: L264-265 Please explain in caption the three numbers what they refer to each time (RMSE 160.5 vs 210.6 vs 217.9). Why are you only showing it for 3 of the 4 models?

**Response**: Agreed. We will include the RMSE for all the four models and make sure it is clear which model each value belongs to.

**Comment**: Figure 2. Can the authors specify the model formula in a and b as well like they do for c and d?

**Response**: We will add this to the panels as suggested.

**Comment**: Can they explain in caption what a, b stands for?

**Response**: Thank you for the suggestion. We will clarify that these are the parameter estimates for the equations.

**Comment**: Can e be shown in the same order as they show the models (ILUA2, Chave, Fitted1, Fitted2)?

**Response**: Good suggestion. We will update the order for the models as suggested.

**Comment:** L266 Please explain how the ILUA2 model was developed more clearly in the methods where introducing this model. If not, it would at least be expected to be in the discussion, considering the underestimation.

**Response**: We explore this in more detail in section 4.1 of the discussion, where we provide an overview of how the ILUA2 model was developed.

**Comment:** L297 More information is needed on the field inventory dataset and measurement protocol to understand what height and rho values were measured (how many replicates, which instrument etc)?

**Response**: As explained above, the measurement protocols are explained in detail in the two references provided, as well as Supplementary section S2 and Table S3. We will make sure to emphasise this when revising the paper.

**Comment**: L302 I think you should rephrase it so it's clear that it's the difference (higher values) that leads to the possible reduction in underestimation, not the fact that they're correlated.

**Response**: Agreed, we will clarify this as suggested.

**Comment:** It would be good throughout this section to reiterate what species-specific vs regional level estimates meant in terms of method. That's really the key difference in the paragraphs here so should be 100% clear for readers.

**Response**: To help the reader navigate the results we will re-iterate here the difference between species-specific vs regional level estimates.

**Comment**: Some of this end section in results already read more like discussion (compounded errors etc).

**Response**: Thank you for this feedback, but our preference would be to retain this section as we feel it provides necessary context for the reader to navigate the results.

**Comment**: Figure 3. Can you make it more visual clear the different scenarios? Eg by showing on the left = measured H all three panels, then middle is Species H, right is Regional H (and similar for top to bottom the differences)? Maybe also by using color codes or size for variation in the RMSE and PE (so we can easily see which ones have smaller or larger errors?)

**Response:** Thank you for the suggestions. We will add labels to the figures to indicate what scenario each row/column corresponds to. We would prefer to avoid using colour coding to grade the RMSE and PE of the models, as we fear this could become confusing.

**Comment**: It isn't mentioned that Bayesian modelling was involved – could this be mentioned in the figure caption of those figures that involved it (?figure 4).

**Response**: The hierarchical Bayesian modelling framework was used to fit the H-D allometries and therefore feeds into the scenarios that vary height estimates. We will clarify in the figure legend which equation was used to estimate height, so that it can be more easily linked back to the methods.

**Comment:** L307-320 This section involved 3 figures as output and the information is not separated clearly enough for readers to digest it. Please try to make this clearer what you found in every step and every figure.

**Response**: Agreed. We will refer more carefully to the figures as 3a, 3b etc. when describing the results.

**Comment**: The abstract should become more clear and concrete when having read the results now. You mention in L30 'using generic information' – please make this specific and linked to how you name the 9 scenarios in the actual article, eg measured, species, regional, ...

**Response**: As mentioned above, we will rephrase this section of the abstract to make our scenarios clearer from the start.

**Discussion**

**Comment:** L385-387 Please explain more carefully why this priority stems from the previous lines of text

Response: This priority stems from the fact that our analysis shows that the pantropical model developed by Chave et al 2014 is actually able to generate robust estimates of AGB as long as reliable values of height and wood density are available. Consequently, we argue that the field should move away from local-scale AGB models fit with small numbers of trees, and instead focus on (1) developing more general AGB models that span a large number of species, trees and tree sizes and therefore provide consistent estimates of AGB, and (2) improving access to robust H-D allometries and wood density values that can serve as inputs to these general AGB models. We will make sure to explain this logic more clearly in revising the paper and provide references to recent efforts to improve access to H-D allometry data (e.g, Tallo database) and comprehensive species-level wood density estimates (e.g., v2 of the Global Wood Density Database which will be released soon).

**Comment:** I miss a section where the authors zoom out on their results and bring forward 'lessons learnt' for people measuring and utilizing numbers on carbon stocks in practice. Generally, the discussion ends on a very specific statement and no any conclusion is made.

**Response:** We have tried to provide conclusions and/or recommendations within each section of the discussion, but agree that summarising these again at the end would be useful for the reader. We will therefore add a short summary section at the end of the discussion to highlight the lessons learned from our study and how this work could be built on in the future.